# Continuity of care interventions for people with stroke and their caregivers: A scoping review protocol

Meera Premnazeer[1], Sarah E. P. Munce[1,2,3], Mark T. Bayley[1,2,3,4], Kristina M. Kokorelias[1,5], Monique Gill[1], Orianna L. Scali[1], Jill I. Cameron[1,2,6]*

**1** Temerty Faculty of Medicine, Rehabilitation Science Institute, University of Toronto, Toronto, Ontario, Canada, **2** KITE, Toronto Rehabilitation Institute, Toronto, Ontario, Canada, **3** Institute of Health Policy, Management and Evaluation, University of Toronto, Toronto, Ontario, Canada, **4** Physical Medicine and Rehabilitation, University of Toronto, Toronto, Ontario, Canada, **5** St. John's Rehab Research Program, Sunnybrook Research Institute, Sunnybrook Health Sciences Centre, Toronto, Ontario, Canada, **6** Department of Occupational Science and Occupational Therapy, Temerty Faculty of Medicine, University of Toronto, Toronto, Ontario, Canada

* jill.cameron@utoronto.ca

## Abstract

### Background

The needs of people with stroke and caregivers' change as they transition across the stroke care continuum from initial symptom onset to community living. Continuity of care interventions may support these changing needs. Continuity of care refers to care provided to individuals that is coherent, connected, and consistent across settings. Interventions have been developed to enhance continuity of care, but to date, no reviews have identified the core components of these interventions specific to various care transitions.

### Objective

To examine the literature on continuity of care interventions that address the needs of people with stroke and/or their caregivers specific to stroke care transitions.

### Methods

The study is guided by the Joanna Briggs Institute methodological framework for scoping reviews. The search will be conducted on Ovid MEDLINE, CINAHL Plus with Full Text, PsycINFO, and EMBASE, restricting to articles in English. Core concepts of stroke (e.g., ischemic or hemorrhagic stroke) and continuity of care (e.g., early supported discharge) and their synonyms will be used as search terms. Two reviewers will assess articles for inclusion. Data will be extracted and synthesised using quantitative descriptive analysis and deductive content analysis. A steering committee,

**Data availability statement:** No datasets were generated or analysed during the current study. All relevant data from this study will be made available upon study completion.

**Funding:** M. Premnazeer is supported by a trainee award from the Rehabilitation Science Research Network for COVID within the Temerty Faculty of Medicine, University of Toronto. M. Premnazeer is also supported through the Ontario Graduate Scholarship, Toronto Rehabilitation Institute Scholarship, and Gwen Bell Endowment Fund from the Rehabilitation Sciences Institute. The funders did not have any role in study design, data collection and analysis, decision to publish, or preparation of the manuscript.

**Competing interests:** The authors have declared that no competing interests exist.

consisting of people with lived experiences, researchers, and healthcare professionals, will discuss the charted data, and identify any gaps in the literature.

## Discussion

This research will identify the components of continuity of care interventions, the specific transitions that they address, and any variability in components across different transitions. This research will identify gaps in the continuity of care literature such as where transition interventions are needed.

---

## Introduction

Stroke is a leading cause of disability in Canada [1]. It can be associated with a myriad of complications, including difficulty with walking, communicating, and a change in cognitive abilities [2,3]. Due to this complexity, people with stroke (PWS) transition between many elements of the health care system, including acute care hospitals, rehabilitation hospitals, and community care services. PWS require support for transitions between care environments.

In addition to the support PWS receive from the formal health care system, they also receive support from family members and friends (i.e., caregivers) who provide unpaid assistance [4]. Caregivers may assist PWS with a variety of daily activities such as cooking or toileting [5]. PWS and caregivers receive limited preparation and supports for transitions across settings (e.g., acute care to home) from the formal health care system due to reduced hospital lengths-of-stay [6] and limited continuity across services [7–9].

The Canadian Stroke Best Practice Recommendations emphasize the importance of supporting PWS and caregivers as they transition across settings [10]. The Canadian Stroke Best Practice Guidelines Transitions of Care Following Stroke Model outlines these transitions (e.g., acute care to rehabilitation) where continuity of care is needed [3]. Continuity of care is coherent, connected, and consistent care provided based on a person's needs and personal context as they navigate different environments [11]. Continuity of care has three dimensions that enhance trust and communication: (1) informational continuity: transfer and use of past medical history to inform present care across providers and settings; (2) management continuity – consistent and timely care is provided to manage a patient's health needs (i.e., shared management plans across providers); and (3) relational continuity – consistent and continuous patient-provider relationship over time and/or across environments [11–13]. This review will operationalize continuity of care as interventions that comprise at least one of the three main dimensions outlined above.

The Canadian Stroke Best Practice Recommendations also outline the importance of providing family-centered care as PWS and caregivers transition [3]. A recent scoping review identified key aspects of family-centered care – consideration of patient, family, and care provider collaboration, family contexts, dedicated policies, and illness-specific education in the development and implementation of patient care

plans [5]. These key aspects were consistent across patient populations, age groups, and care contexts leading to the development of the Universal Model of Family-Centered Care. This model is central to supporting both patients and their caregivers.

Many reviews on continuity of care interventions and stroke have been published [14–17]. A recent meta-analysis found that continuity of care interventions improved stroke patients' independence compared to usual care [15]. A systematic review found low-to-moderate strength of evidence for the effectiveness of hospital-initiated transitional care for PWS or myocardial infarction [16]. Another systematic review found support interventions to home after acute stroke improved functional status and quality of life in the short-term, and depression and anxiety of PWS in the medium to long-term [14]. A literature review examining the effectiveness of continuity of care interventions for hospital discharge to the home found some interventions reported positive outcomes (e.g., quality of life, depression and mood) [17]. A recent review emphasized the importance of person-centered care in continuity of care interventions [18].

Previous reviews did not aim to comprehensively review all transitions across the care continuum (e.g., stroke onset to the emergency department, inpatient acute care to inpatient rehabilitation). For example, O'Callaghan et al. (2022) only examined the single transition from hospital to home or inpatient settings to home. PWS and caregivers enter the healthcare system much earlier and require continuity of care support from stroke symptom onset as identified by the Canadian Best Practice Recommendations [3].

Reviews to date [14–18] also focused on person-centered care and rarely considered family-centered care (e.g., providing caregivers with illness-specific education) [16]. An understanding of transition-specific and stroke-specific needs over time is needed to inform the development of future continuity of care interventions, as PWS and caregivers have differing needs and supports as they move across the care continuum [19].

To date, no reviews have characterized continuity of care interventions for PWS and their caregivers across the entire stroke care continuum and no reviews have explored the provision of family-centered care, specifically. The main objective of this scoping review is to examine the literature on continuity of care interventions that address the needs of PWS and/or their caregivers as they transition. Specific objectives of the scoping review are to:

1. Map the transitions points in the continuum of care that continuity of care interventions address (i.e., transitions found within the Canadian Stroke Best Practice Guidelines Transitions of Care Following Stroke Model [3]).

2. Summarize continuity of care models based on the components identified within the Template for Intervention Description and Replication (TIDieR) checklist [20].

3. Summarize and characterize the extent to which domains of continuity of care and family-centered care are included in continuity of care interventions.

A preliminary search for existing reviews on the topic of continuity of care and stroke was conducted on Joanna Briggs Institute (JBI) Database of Systematic Reviews and Implementation Reports, the Cochrane Database of Systematic Reviews, Open Science Framework and MEDLINE in December 2022, February 2023, October 2023, and June 2024. No reviews were identified that met the primary aim of this study. The review has been registered within Open Science Framework (10.17605/OSF.IO/J6H4R). Amendments to the review will be reported within the completed scoping review.

## Methods

### Design

The scoping review will be conducted following the JBI methodology for Scoping Reviews [21,22]. Reporting will follow the Preferred Reporting Items for Systematic Reviews and Meta-Analyses extension for Scoping Reviews (PRISMA-ScR) [23] and Preferred Reporting Items for Systematic Review and Meta-Analysis Protocols (PRISMA-P) [24].

## Steering committee

A steering committee includes members of the investigative team including two people with lived experience (PWS and caregiver). The investigative team includes individuals who have expertise in stroke, caregiver, and care continuum research [3,25–27]. The team also consists of individuals who have experience in conducting scoping reviews pertaining to PFCC [5,28–30].

The study objectives and study protocol have been reviewed by experts in the field of stroke (e.g., healthcare professionals who work with PWS and researchers). The steering committee will review, refine, and finalize the research objectives, methodological approaches, results, and dissemination strategies.

## Eligibility criteria

**Participants.** The population of interest includes people with all types of strokes and/or their caregivers (any age). Included studies must have either PWS or their caregivers as participants. Caregivers will be defined as family members or friends who assist with at least one basic (e.g., dressing, toileting) or instrumental (e.g., banking, grocery shopping) activity of daily living per week [31].

**Concept.** The main concepts of the article will be continuity of care interventions (e.g., the Timing it Right Stroke Family Support Program [27]) for the stroke population. The article should examine at least one transition identified in the Canadian Stroke Best Practice Guidelines Transitions of Care Following Stroke Model (e.g., moving from initial onset to acute care or from acute care to community, long-term care, and palliative care) [3,10,25,32]. However, studies that focus on continuity of care interventions for healthcare professionals training to address patient and caregiver needs will be excluded.

**Context.** Articles describing studies conducted anywhere in the world will be included. The included articles must be written in English as we did not have resources to support translation. This is a limitation of the study as it may result in the exclusion of relevant articles in other languages. The search criteria will include articles published between January 2000 to present. The restriction to published literature on continuity of care interventions from the year 2000 onwards is to maintain relevance to current practices and ensure consistency with previous reviews [15–17].

**Types of evidence sources.** The review will only include randomized control trials. This was chosen to: 1) ensure feasibility of the review as we identified a large number of continuity of care interventions during our initial searches and 2) include high-quality evidence. This is a limitation of the study as it may exclude interventions not yet tested at the randomized control trial level, potentially affecting the comprehensiveness of mapping the interventions to date. Specifically, the review will exclude non-randomized controlled trials, quasi-experimental studies, case reports, descriptive cross-sectional studies, and any other experimental and observational quantitative study designs. The review will also exclude qualitative study designs such as qualitative descriptive, grounded theory, and ethnographic designs. Review articles such as scoping reviews, systematic reviews, and meta-analysis will be used to identify other potential original articles to include in the review that meet the inclusion criteria. Opinion pieces, grey literature, unpublished studies, viewpoints, commentaries, dissertations, theses, book chapters, books, news articles, letters, blogs, editorials, reports, protocol papers and abstracts will not be included.

## Search strategy

The research team has developed the search strategy and have consulted a librarian affiliated with the Rehabilitation Sciences Institute at the University of Toronto (See attached search strategy in S1 Appendix A for Ovid MEDLINE Search). Consensus meetings have been held to refine the search strategy. The search strategy will be adapted for each database. The search terms utilized will be consistent with previous reviews conducted on this topic [5,17]. Preliminary searches of Ovid MEDLINE and PsycINFO were conducted to identify additional key search terms commonly found within titles, abstracts, and index terms of relevant studies.

Four electronic databases will be used: (1) Ovid MEDLINE, (2) CINAHL Plus with Full Text, (3) PsycINFO, (4) and EMBASE. These databases were selected as they have previously been used to conduct scoping reviews on continuity of care interventions [5,11,17,33]. The research team will examine the reference lists of included articles and previous literature reviews to identify and include articles that meet the inclusion criteria. This method has been known to assist with identifying additional references [11].

## Study/source of evidence selection

Following the database search, all identified citations will be imported into EndNote and then Covidence where duplicates will be removed [34]. Covidence is a platform that supports screening, reviewing articles, and charting data and has been used previously for scoping reviews [34].

First, a pilot test of the screening process will be conducted with a small subset of articles. Two independent reviewers will screen the title and abstract of 20 articles against the inclusion and exclusion criteria outlined above. The inclusion and exclusion criteria will be modified to enhance clarity between the two reviewers as needed. The pilot test will be repeated in batches of 20 until a Cohen's kappa of 0.80 is achieved. This cut-off has been used in other reviews [35–37]. Once a Cohen's kappa of 0.80 is achieved between the two reviewers, the title and abstract screening of the remaining articles will commence.

The same pilot test will be conducted for full-text screening with 20 articles, and it will be repeated if necessary. This will be completed by the same two reviewers and inclusion criteria will be clarified. Full-text screening of the remaining articles will be conducted once a Cohen's kappa of 0.80 is achieved. Reason for exclusion during full-text screening will be recorded and reported in the PRISMA flow diagram of the scoping review. Discrepancies between reviewers during title and abstract or full-text screening will be addressed by a third reviewer or through consensus meetings, if necessary.

As this is a scoping review and not a systematic review, a risk of bias of included studies will not be conducted as it is not a mandatory step and is not often done [38].

## Data extraction

Data will be extracted by two or more independent reviewers using a data extraction tool on Covidence. A pilot test of the full data extraction tool will be completed with a random selection of at least 10 articles. The research team will discuss discrepancies in data extracted and will modify the data extraction template to enhance consistencies across reviewers. Authors of papers that are included may be contacted to request missing or additional data.

The following descriptive data will be extracted from articles and presented in a summary table [39]:

- Author, year of publication, Name of Journal

- Care contexts (setting)

- Country

- Objective and Question(s) of the study

- Study design

- Study population

- Sample size

- Sex and/or gender

- Specific transition (e.g., acute care to home)

- TIDieR checklist describing the interventions [20]

- Components of the intervention as they relate to family-centered care and continuity of care frameworks

The draft extraction tool may be updated as the review progresses. The modifications will be outlined in the scoping review publication.

## Data analysis and presentation

The scoping review will follow the JBI guidance (2022) to engage in data analysis and presentation through: (1) providing a descriptive numerical summary of the data and deductive content analysis (2) report results by relating to objectives and research questions, and (3) relate the overall findings to implications they have for stroke care [40].

Descriptive numerical summary will be completed using Microsoft Excel [21]. This data will summarize the characteristics of included studies using frequencies and percentages (e.g., overall number of studies included, percentage of interventions that address specific transitions) [41].

A deductive content analysis will be used to synthesize qualitative data [40,42,43]. The review authors will familiarize themselves with the data and extract data according to the main objectives: (1) Mapping continuity of care interventions based on specific transitions across settings (based on Canadian Stroke Best Practice Guidelines Transitions of Care Following Stroke Model [3]); (2) Summarizing components of continuity of care models developed for individual transitions (TIDieR); and (3) Characterizing the extent to which domains of continuity of care and family-centered care are addressed by the interventions. Extracted data will be reviewed by members of the review team to ensure that the information obtained reflects the understanding of the frameworks. The data will be presented as a narrative summary that gives an overall explanation of the findings in relation to the objectives [44].

## Co-creation exercise

Pollock et al. (2022) outline the co-creation exercise as an essential step in the scoping review process for health-related issues. Involving relevant knowledge users in the final review can enhance the findings of the overall review [39,45]. This step will also help to integrate knowledge that may not be accessible within the literature (e.g., new trends that have not been published or historical contexts), increase transparency and rigour, and improve dissemination of findings [45,46]. A virtual meeting will be organized with the steering committee (i.e., investigators, clinicians, program managers, knowledge users in stroke program development and administration, and people with lived experience). This meeting will discuss emerging themes and identify needs for future practice and research pertaining to stroke care (see S3 Appendix B for interview guide informed from a study on developing questions for focus groups [47]). This co-creation exercise will also be a unique addition as previous scoping reviews of this and related topics did not include this step. This step will also aid in expanding the outreach and dissemination of findings to various knowledge users. Contributions within the co-creation process may provide additional references to be reviewed and identify any gaps in the scoping review [39]. Feedback provided during the meeting will be summarized, incorporated into the qualitative thematic analysis and integrated in the overall study outcome [48]. Knowledge users who participated in the virtual meeting will be invited to share the findings from the scoping review to the wider community at the end of the study [45].

## Ethical considerations

Institutional Research Ethics Boards at the University of Toronto will not need to review or approve our co-creation exercise as the members will be part of our existing steering committee.

## Conclusion

Scoping reviews are identified as one method to inform best practice models [49]. This scoping review will be an effective tool to map continuity of care interventions across the stroke care continuum (e.g., acute care to rehab) and is further strengthened by the co-creation exercise.

To date, most reviews on this topic focus on the transition from hospital to home. The purpose of this scoping review is to examine all transitions across the stroke care continuum. This is important as this research will identify gaps in the continuity of care literature such as where in the care continuum continuity of care interventions have not been studied. This will provide recommendations for future research.

This research will identify and summarize components of continuity of care models that are specific to individual stroke transitions across the entire stroke care continuum as outlined by the Canadian Best Practices Stroke Guidelines Transitions of Care Following Stroke Model [3]. As studies from across the world will be included, transitions points not identified by the model will be outlined to inform future models of the stroke care continuum. The review will also summarize and characterize the extent to which domains of continuity of care and family-centered care are addressed in the continuity of care models. Changes that will be made to the scoping review protocol will be reported in the final publication.

The findings from this review will be shared with relevant knowledge users within the steering committee which include developers of the Canadian Best Practices Stroke Guidelines. Findings will also be shared through conferences presentations. The results of the scoping review will be submitted for publication in a scientific journal.

## Supporting information

**S1 Appendix. This is the Search Strategy for MEDLINE via Ovid.**
(DOCX)

**S2 Appendix. This is the Draft Interview Guide for the Co-Creation Exercise.**
(DOCX)

**S3 Appendix. This is the PRISMA-P 2015 Checklist.**
(DOCX)

## Acknowledgments

We would like to thank, Katie Merriman, a librarian at the University of Toronto, for their support and consultation in the search strategy for the study.

## Author contributions

**Conceptualization:** Meera Premnazeer, Sarah E. P. Munce, Mark T. Bayley, Jill I Cameron.

**Data curation:** Meera Premnazeer, Kristina M. Kokorelias, Monique Gill, Orianna L. Scali, Jill I Cameron.

**Funding acquisition:** Meera Premnazeer.

**Investigation:** Meera Premnazeer.

**Methodology:** Meera Premnazeer, Sarah E. P. Munce, Mark T. Bayley, Monique Gill, Jill I Cameron.

**Project administration:** Meera Premnazeer.

**Supervision:** Sarah E. P. Munce, Mark T. Bayley, Jill I Cameron.

**Writing – original draft:** Meera Premnazeer, Jill I Cameron.

**Writing – review & editing:** Meera Premnazeer, Sarah E. P. Munce, Mark T. Bayley, Kristina M. Kokorelias, Monique Gill, Orianna L. Scali, Jill I Cameron.

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
