## [Decision Letter · Decision Letter 0]

5 Nov 2024

PONE-D-24-30063Continuity of Care Interventions for People with Stroke and their Caregivers: A Scoping Review ProtocolPLOS ONE

Dear Dr. Cameron,

Thank you for submitting your manuscript to PLOS ONE. After careful consideration, we feel that it has merit but does not fully meet PLOS ONE’s publication criteria as it currently stands. Therefore, we invite you to submit a revised version of the manuscript that addresses the points raised during the review process.

We look forward to receiving your revised manuscript.

Kind regards,

Dr Tazeen Majeed

Academic Editor

PLOS ONE

Journal Requirements:

Reviewers' comments:

Reviewer's Responses to Questions

**Comments to the Author**

1. Does the manuscript provide a valid rationale for the proposed study, with clearly identified and justified research questions?

Reviewer #1: Yes

Reviewer #2: Yes

2. Is the protocol technically sound and planned in a manner that will lead to a meaningful outcome and allow testing the stated hypotheses?

Reviewer #1: Yes

Reviewer #2: Yes

3. Is the methodology feasible and described in sufficient detail to allow the work to be replicable?

Reviewer #1: Yes

Reviewer #2: Yes

4. Have the authors described where all data underlying the findings will be made available when the study is complete?

Reviewer #1: Yes

Reviewer #2: Yes

5. Is the manuscript presented in an intelligible fashion and written in standard English?

Reviewer #1: Yes

Reviewer #2: Yes

6. Review Comments to the Author

You may also provide optional suggestions and comments to authors that they might find helpful in planning their study.

Reviewer #1: The protocol is well-written and addresses a good clinical question but has minor corrections to be made.

Reviewer #2: The manuscript presents an exceptionally well-organized and thorough study protocol that addresses an important gap in stroke care research, specifically in continuity of care interventions for people with stroke (PWS) and their caregivers. The work demonstrates a deep understanding of both the clinical and research landscapes, and the methodology is meticulously designed to ensure a comprehensive scoping review. The authors have successfully articulated the relevance of their study, which will likely contribute significantly to improving stroke care transitions.

However, there are some areas for improvement that could enhance the clarity and impact of the manuscript. In the abstract, while the objectives and methods are clearly stated, further detail could be added to the methods section to outline the databases and search terms more explicitly. This would provide readers with a clearer understanding of the scope of the research at a glance. The introduction is comprehensive but could benefit from being more concise, with a more direct focus on the specific research gap. Some redundancy in the background could be streamlined to improve flow and reader engagement. In the methodology, the choice to limit the review to randomized controlled trials could be reconsidered, as scoping reviews typically aim to include a broader range of evidence types. Expanding the types of studies included would align more closely with the objectives of mapping the breadth of available evidence on continuity of care interventions.

Overall, the manuscript showcases a high level of scholarly rigor and has great potential for advancing research in stroke care. With these small adjustments, the work can be further strengthened to ensure maximum impact and clarity.

7. PLOS authors have the option to publish the peer review history of their article (what does this mean? ). If published, this will include your full peer review and any attached files.

**Do you want your identity to be public for this peer review?** For information about this choice, including consent withdrawal, please see our Privacy Policy .

Reviewer #1: No

Reviewer #2: **Yes: ** Salman Ashfaq Ahmad

---

## [Author Response · Author response to Decision Letter 0]

21 Nov 2024

Dear Editor,

Thank you for giving us an opportunity to submit a revised draft of our manuscript “Continuity of Care Interventions for People with Stroke and their Caregivers: A Scoping Review Protocol”. We have incorporated the suggestions made by the reviewers. All line and page numbers refer to the revised manuscript file with tracked changes. We have additionally changed the following: (1) added initials for all authors (Line 5-6) and (2) the ordering in our acknowledgement section has been rearranged to place importance on funding most relevant to the current study (Line 328-332). All page numbers refer to the revised manuscript with tracked changes.

Reviewer’s Comments to the Authors:

Reviewer 2: In the abstract, while the objectives and methods are clearly stated, further detail could be added to the methods section to outline the databases and search terms more explicitly. This would provide readers with a clearer understanding of the scope of the research at a glance.

Author response: Thank you for the feedback. The databases that will be used are outlined in Line 10-11 (Ovid MEDLINE, CINAHL Plus with Full Text, PsycINFO, and EMBASE). We have added additional examples for search terms beside the main concepts of stroke and continuity of care (Line 12-13).

Specific Change: “The search will be conducted on Ovid MEDLINE, CINAHL Plus with Full Text, PsycINFO, and EMBASE, restricting to articles in English. Core concepts of stroke (e.g., ischemic or hemorrhagic stroke) and continuity of care (e.g., early supported discharge) and their synonyms will be used as search terms.”

Page Number: 2

Reviewer 2: The introduction is comprehensive but could benefit from being more concise, with a more direct focus on the specific research gap. Some redundancy in the background could be streamlined to improve flow and reader engagement.

Author response: Thank you for the suggestion. We have edited the introduction such that it is more concise and avoids redundancy.

Specific Change: Removed redundant information (Line 27, 33, 74-75)

Changed phrasing on Line 39 “move from one environment to another” to “navigate different environments.”

Simplified section on continuity of care and its dimensions to: “Continuity of care has three dimensions that enhance trust and communication: (1) informational continuity: transfer and use of past medical history to inform present care across providers and settings; (2) management continuity – consistent and timely care is provided to manage a patient’s health needs (i.e., shared management plans across providers); and (3) relational continuity – consistent and continuous patient-provider relationship over time and/or across environments (12-14). This review will operationalize continuity of care as interventions that comprise at least one of the three main dimensions outlined above.” (Line 39-44)

Simplified the summary of continuity of care interventions paragraphs (Lines 82, 85, 87, 89-90, 93-113, 117, 120): “Many reviews on continuity of care interventions and stroke have been published (15-18). A recent meta-analysis found that continuity of care interventions improved stroke patients’ independence compared to usual care (16). A systematic review found low-to-moderate strength of evidence for the effectiveness of hospital-initiated transitional care for PWS or myocardial infarction (17). Another systematic review found support interventions to home after acute stroke improved functional status and quality of life in the short-term, and depression and anxiety of PWS in the medium to long-term (15). A literature review examining the effectiveness of continuity of care interventions for hospital discharge to the home found some interventions reported positive outcomes (e.g., quality of life, depression and mood) (18). A recent review emphasized the importance of person-centered care in continuity of care interventions (19).

Previous reviews did not aim to comprehensively review all transitions across the care continuum (e.g., stroke onset to the emergency department, inpatient acute care to inpatient rehabilitation). For example, O’Callaghan et al. (2022) only examined the single transition from hospital to home or inpatient settings to home. PWS and caregivers enter the healthcare system much earlier and require continuity of care support from stroke symptom onset as identified by the Canadian Best Practice Recommendations (3).

Reviews to date (15-19) also focused on person-centered care and rarely considered family-centered care (e.g., providing caregivers with illness-specific education) (17). An understanding of transition-specific and stroke-specific needs over time is needed to inform the development of future continuity of care interventions, as PWS and caregivers have differing needs and supports as they move across the care continuum (20).”

Page Number: 3-5

Reviewer 2: In the methodology, the choice to limit the review to randomized controlled trials could be reconsidered, as scoping reviews typically aim to include a broader range of evidence types. Expanding the types of studies included would align more closely with the objectives of mapping the breadth of available evidence on continuity of care interventions.

Author response: Thank you for your feedback regarding our choice to limit the review to randomized controlled trials (RCTs). We agree that expanding the type of studies would align more closely with the objectives of mapping the breadth of available evidence on continuity of care interventions. However, our initial searches showed a large number of continuity of care interventions in this area (100+). Thus, to ensure feasibility we decided to focus on RCTs while still capturing high-quality evidence. We also acknowledge this as a limitation (Line 192-195).

Specific Change: “The review will only include randomized control trials. This was chosen to: 1) ensure feasibility of the review as we identified a large number of continuity of care interventions during our initial searches and 2) include high-quality evidence. This is a limitation of the study as it may exclude interventions not yet tested at the randomized control trial level, potentially affecting the comprehensiveness of mapping the interventions to date.”

Page Number: 8

Reviewer: This is the first time this abbreviation is being mentioned, therefore the authors need to state the full meaning.

Author response: Thank you for outlining this. We have expanded the abbreviation for PWS to “people with stroke” (Line 8).

Specific Change: “To examine the literature on continuity of care interventions that address the needs of people with stroke and/or their caregivers specific to stroke care transitions.”

Page Number: 2

Reviewer: The Joanna Briggs Institute Methodological Framework refers to a specific methodology for reviewing articles. However, the authors did not specify the exact methodology used in this study.

Author response: Thank you for this feedback. We have added within the sentence “scoping review” to indicate the specific methodology that the Joanna Briggs Institute Methodological Framework refers to (Line 10). This was also added within the Methods (Line 154-155).

Specific Change: “The study is guided by the Joanna Briggs Institute methodological framework for scoping reviews.”

“The scoping review will be conducted following the JBI methodology for Scoping Reviews (22, 23).”

Page Number: 2, 6

Reviewer: The authors have not stated why they limited their search to English language studies only. Additionally, they had earlier stated that they will include studies conducted anywhere in the world.

Author response: Thank you for pointing this out. This review was limited to English as we did not have access to or resources to support translation of articles from other languages to English. We mention this as a limitation (Line 184-186).

Specific Change: “The included articles must be written in English as we did not have resources to support translation. This is a limitation of the study as it may result in the exclusion of relevant articles in other languages.”

Page Number: 7

---

## [Decision Letter · Decision Letter 1]

8 Apr 2025

Continuity of Care Interventions for People with Stroke and their Caregivers: A Scoping Review Protocol

PONE-D-24-30063R1

Dear Dr. Cameron,

We’re pleased to inform you that your manuscript has been judged scientifically suitable for publication and will be formally accepted for publication once it meets all outstanding technical requirements.

Kind regards,

Marianne Clemence

Staff Editor

PLOS ONE

Additional Editor Comments (optional):

Reviewers' comments:

Reviewer's Responses to Questions

**Comments to the Author**

1. Does the manuscript provide a valid rationale for the proposed study, with clearly identified and justified research questions?

Reviewer #1: Yes

Reviewer #2: Yes

2. Is the protocol technically sound and planned in a manner that will lead to a meaningful outcome and allow testing the stated hypotheses?

Reviewer #1: Yes

Reviewer #2: Yes

3. Is the methodology feasible and described in sufficient detail to allow the work to be replicable?

Reviewer #1: Yes

Reviewer #2: Yes

4. Have the authors described where all data underlying the findings will be made available when the study is complete?

Reviewer #1: Yes

Reviewer #2: Yes

5. Is the manuscript presented in an intelligible fashion and written in standard English?

Reviewer #1: Yes

Reviewer #2: Yes

6. Review Comments to the Author

You may also provide optional suggestions and comments to authors that they might find helpful in planning their study.

Reviewer #1: I think the authors have a good job by responding well to the concerns that were initially raised. I would recommend accepting the article for publication. Thank you

Reviewer #2: The author of the study protocol has addressed all the comments. The study protocol is now up to the standards and can be published.

7. PLOS authors have the option to publish the peer review history of their article (what does this mean? ). If published, this will include your full peer review and any attached files.

**Do you want your identity to be public for this peer review?** For information about this choice, including consent withdrawal, please see our Privacy Policy .

Reviewer #1: **Yes: ** Dr Collins Ogbeivor

Reviewer #2: **Yes: ** Salman Ashfaq Ahmad

---

## [Editor Report · Acceptance letter]

PONE-D-24-30063R1

PLOS ONE

Dear Dr. Cameron,

I'm pleased to inform you that your manuscript has been deemed suitable for publication in PLOS ONE. Congratulations! Your manuscript is now being handed over to our production team.

Kind regards,

on behalf of

Dr Marianne Clemence

Staff Editor

PLOS ONE